# Laser Welding of Ti6Al4V Titanium Alloy in Air and a Water Medium

**DOI:** 10.3390/ma15249088

**Published:** 2022-12-19

**Authors:** Mohamad Alhajhamoud, Sayit Ozbey, Mehmet Alp Ilgaz, Levent Candan, Ibrahim Cinar, Mario Vukotić, Selma Čorović, Damijan Miljavec, Ersin Kayahan

**Affiliations:** 1Biomedical Engineering, Natural and Applied Sciences, Kocaeli University, Umuttepe, Kocaeli 41380, Turkey; 2Laser Technologies Research and Application Center (LATARUM), Kocaeli University, Yeniköy, Kocaeli 41275, Turkey; 3Maritime Faculty, Kocaeli University, Marine Eng. Karamürsel, Kocaeli 41500, Turkey; 4Faculty of Electrical Engineering, University of Ljubljana, 1000 Ljubljana, Slovenia

**Keywords:** laser welding, Ti6Al4V titanium, Nd:YAG laser, depth of penetration, laser material interaction

## Abstract

Ti6Al4V titanium alloys are widely used in a variety of scientific and industrial fields. Laser beam welding is one of the most effective techniques for the joining of titanium plates. The main objective of this study was to investigate the influence of the most important laser parameters on welding performance of titanium alloy in two different physical environments such as air and water (i.e., serum) media. Specifically, the laser beam welding of 2 mm thick Ti6Al4V samples was applied using an Nd:YAG laser in open-air welding using argon as a shielding gas, and in wet welding using a serum environment. The deepest penetration was achieved at −3 mm focal position with 11 J of laser energy in both investigated media (i.e., air and serum). The maximum hardness (1130 HV) was achieved for the focal position of −4 mm in serum medium while it was 795 HV for a focal position of −5 mm in air medium. The minimum (1200 μm and 800 μm) and maximum (1960 μm and 1900 μm) weld widths were observed for air and serum medium, respectively. After the welding process, martensite, massif martensite, and transformed martensite were observed in the microstructure of Ti6Al4V. To the best of our knowledge, the underwater wet welding of titanium alloy was carried out and reported for the first time in this study.

## 1. Introduction

Ti6Al4V titanium alloys are commonly used in automotive, aviation, biomedical, aerospace, petrochemical, ship building, nuclear, and power generation industries because of their excellent properties such as high strength, low density, high fracture toughness, fatigue strength, resistance to crack propagation, superior biocompatibility, and superior corrosion resistance [1,2,3,4,5,6]. Ti6Al4V is also preferred as a prosthetic material [7] due to its high biocompatibility [8] and remarkably low corrosion rates in body fluids [9]. Ti6Al4V, which has been preferred by a wide range of industries for half a century, has proven to be technically superior and cost-effective [10,11]. Additionally, compared to other metal alloys, its manufacturability, weldability, and workability are far superior [12]. Ti6Al4V titanium alloys, which are corrosion resistant, are also used in the maritime and nuclear industries, where structural components are directly exposed to water [13].

Underwater welding (UW) has become the most extensively utilized method of repair of offshore pipeline platforms, boats, seashore components, and port equipment [14]. In situ joining and repair of underwater operations can be performed with underwater laser welding. It is a suitable repair method for large-scale constructions since it considerably reduces the cost of part disassembly and transportation. UW can be classified into three main groups: underwater dry welding (UDW), underwater wet welding (UWW), and underwater local cavity welding (ULDCW) [14,15]. UDW is typically used in underwater high-pressure chambers and can produce high-quality welded connections, although its welding equipment is more sophisticated and expensive compared to the UWW [16]. The important advantage of UWW is that it can prevent fractures and corrosion [17]. On the other hand, during UWW, water vapor, diffusible hydrogen, or carbon monoxide may create porosity [18]. ULDCW needs to be performed in a controlled environment and a pressure chamber. Compared to UWW, ULDW produces higher-quality outcomes but requires more sophisticated equipment and higher costs [15]. Wet welding is preferred over other repair methods since it can be operated with ordinary and cost-effective equipment rapidly, minimizing repair costs and downtime [19]. In addition to the techniques mentioned above, the arc welding approach can also be used as an effective welding method. However, arc welding may result in larger heat affected zone (HAZ) and weld width, which is not acceptable for the welding process of the titanium alloys.

With recent advancements in welding technology, a variety of joining technics for titanium alloys have been developed to enhance weld quality, efficiency, cost-effectiveness, and safety [19]. However, these materials are sensitive to gases and this restricts the usage of some welding techniques and requires taking precautions [20]. The most widely used welding processes for bonding titanium and its alloys are gas tungsten arc welding (GTAW), gas metal arc welding (GMAW), plasma-arc welding (PAW), laser beam welding (LBW), and electron-beam welding [21,22]. Welding seams with a high aspect ratio can be obtained with or without the use of filler material during the welding process [7]. Compared to GTAW or PAW, the LBW process increases the depth of penetration and reduces possible welding defects and the heat affected zone (HAZ), thus increasing the mechanical performance of welded metal (WM) [23] by enabling the generation of a keyhole that focuses the energy input into a narrow zone [24]. Distortion is less in materials bonded with the LBW method [25,26]. Furthermore, the LBW method’s automation susceptibility provides significant benefits in mass production [27].

Therefore, in this study, the laser welding technique (LBW) has been used since the characteristics of titanium alloys impose strict requirements on the welding procedure.

Laser welding could be performed both underwater and in the open air [28]. This welding technique results in a low HAZ and residual stress, while maintaining a high welding quality [29]. Therefore, underwater laser welding technology offers a lot of potential for future welding and repair of deep, large-scale structures [30].

Temperature irregularity, non-uniform chemical composition, and stress make LBW of aluminum alloys a complex process. The microstructure and mechanical properties of titanium alloys are affected by heat input because of poor conductivity [31]. In addition to these obstacles in air welding of titanium, additional problems arise in the UW process. In traditional underwater arc welding, the high-pressure water environment imposes challenges such as hydrogen, cracking, and residual stresses [32]. Weldability tests of S460N high strength steel were carried out in a water environment, indicating that the welds were sensitive to cold cracking due to their high hardness [33]. The influence of welding parameters and circumstances on the diffusible hydrogen content in underwater deposited metal was explored, and it was reported that decreasing the welding speed reduces the diffusible hydrogen content [34]. An extensive study on pore formation in underwater wet welds confirmed that hydrogen and/or vapors are responsible for the pore formation [18]. It has also been reported that underwater laser welding (ULW) has several benefits over underwater arc welding, including accurate heat input, decreased residual stress, and higher welding stability [35].

Ti6Al4V is a widely used two-phase titanium alloy (α phase and β phase) and LBW has been frequently used to efficiently connect Ti6Al4V alloys due to its high efficiency, deformation resistance, and welding stability [36]. In addition, the low heat conductivity and high laser absorptivity properties of titanium alloy enable obtention of a high aspect ratio during LBW [37].

A large body of scientific literature predominantly focusses on the welding of titanium alloys in open air medium [3,5,6,38,39]; however, the wet laser welding of Ti6Al4V alloys has not been reported previously. It should be therefore noted that the underwater wet welding of titanium alloy (Ti6Al4V) has been conducted, and the results are reported for the first time in the present study.

In this study, the laser welding of Ti6Al4V alloys was performed with a millisecond pulsed Nd:YAG laser in air and in a biological medium (i.e., serum). The investigated laser beam welding parameters were the focal position and the laser power. The quality of the welding connections was investigated by detailed examination of the microstructures and a hardness test. Additionally, specific point energy (SPE), which includes the parameters of welding speed, focal position, and laser power, was evaluated to find the optimal welding parameters. All parameters used in the experiments were evaluated with respect to the welding environment physical conditions. The possibility of the use of wet welding of titanium alloys in serum medium, and the effect of the medium in which the laser welding has been performed on the welding quality have been discussed in detail.

## 2. Materials and Methods

Experimental measurements have been performed to examine the influence of important laser parameters on the welding quality of the titanium alloy. In the experimental procedure, Ti6Al4V plates with the dimensions of 20 mm × 20 mm × 2 mm were used. First, the prepared samples were carefully cleaned in an ultrasonic bath. After the cleaning process, laser welding of the samples was applied. After the welding process was completed, wire erosion was used to cut small metal samples. The test samples were grinded with 240, 320, 400, 600, 1000, 1200, and 2000 grit SiC sandpaper, polished with diamond polishing paste, cleaned with alcohol, and etched with Kroll’s solution (water (H_2_O) 92.82% + nitric acid (HNO_3_) 6.11% + hydrofluoric acid (HF) 1.07%).

The chemical composition and the physical and mechanical properties of Ti6Al4V are specified in Table 1. The experimentally examined laser parameters are given in Table 2. These parameters have been chosen based on the pre-experimental studies and were investigated in detail, since they have noticeable influence on the welding performance and the SPE value.

The welding process was carried out using an Nd:YAG laser (1064 nm) with maximum output peak power of 10 kW and pulse duration of 0.5–10 ms. The schematic illustration of the experimental setup is given in Figure 1. The welding process was at room temperature. The laser power and focal position have been examined in detail, since they are the most critical parameters influencing the welding quality [41].

The welding process was implemented in two media: atmospheric air and serum. The serum liquid, which was purchased from the Biofleks OSEL company (Beykoz, Istanbul, Turkey) consisted of 99.1% of water, 0.9% of salt, 0.154 mole Na, and 0.154 mole Cl per litre. The serum medium was selected to investigate the feasibility of laser welding inside the human body. Argon gas (18 L/min) was chosen as a protective gas for laser welding in atmospheric air to prevent oxidation and improve joint performance [7].

The SPE can be used to determine the influence of laser parameters. The notion of SPE is utilized in laser welding to determine the range of the penetration [42]. The SPE was evaluated by the mathematical equation 1 [42]:(1)$$Specific\;Point\;Energy\;\left(SPE\right)\left(J\right)=P\frac{D}{V}$$
where *P* is the laser power (W), *D* is the laser beam diameter (mm), and *V* is the welding speed (mm/s). The SPE calculation can be used to estimate the welding capabilities of different laser systems.

The microstructure was observed with an optical microscope (OLYMPUS BX51 M), whereas the variations in hardness between the welded area and the base metal (BM) zone was measured by using Vickers method (HV 0.3) [43].

## 3. Results and Discussion

The welding process was carried out using a solid-state laser on titanium samples in two distinct media (water and air), and the results were compared in order to determine the optimal laser parameters. The most significant parameters that may influence the laser welding performance, such as laser power and focal position, have been examined in both media. In order to achieve a deeper penetration during laser welding, various methods have been proposed in the literature [44,45]. A higher penetration depth can be achieved by increasing the laser power (and/or increasing the laser power to welding speed ratio), by reducing the laser diameter, or by applying the pre-heating weldment [46,47].

Figure 2 shows the macrographs of the specimens carried out in the serum and air medium, while Figure 3 shows the photograph of the experimental setup, welded plate, and samples from the welded plate.

Figure 4 and Figure 5 show SEM and EDS images of the Ti6Al4V samples welded in air (a) and in serum (b), respectively. The SEM images indicate that the surface welded in the air is rougher compared to the surface welded in serum. This can be explained by the fact that the argon gas used as a shielding gas during the laser welding affects the surface properties of the welded region. Moreover, when the welding process is performed in a serum medium, the welded surface oxidizes due to properties of the water environment, as seen in Figure 4b.

In the SEM micrographs (Figure 4a), the samples resemble a sea wave since argon is used as a shielding gas in the welding. The micro image in Figure 4b indicates that elements such as sodium, potassium, etc., were separated in the serum medium. The segment that resembles a round stone represents oxygen, see Figure 4b.

Figure 6 represents the microstructure of BM at room temperature. BM mainly consists of equiaxed α phase which appears light in color and an inter-granular phase which appears dark in color.

Figure 7 demonstrates the microstructures of the underwater- and air-welded metal fusing zones. The columnar crystals were first formed on the BM in the fusion line, and then they progressively developed to the center of the WM. The underwater- and air-welded metals contain acicular martensite at the fusion zones. In the underwater-welded metals, the martensite was mostly distributed in a shape forming intersecting perpendicular structures (‘’basket’’). However, in the air-welded metals, they were distributed randomly. This is because the cooling rate in the underwater environment is greater than the cooling rate in the air environment.

The microstructure of the HAZ of the underwater- and air-WM is shown in Figure 8. In both welding environments, the HAZ could be categorized into two distinct areas: near-weld area and the BM zone. The area near the welding zone is composed of the intersecting perpendicular “basket” samples of α +$\alpha^{\prime }$ phase, while the area near the BM consists of α and β phase.

The laser parameters, such as the focal position and the laser power, have a solid impact on the welding quality. The SPE value can be used to investigate the total impact of laser parameters. The influence of the focal position was examined by varying it from −6 mm to −3 mm while other parameters were kept constant (5 mm/s speed, 11 J energy, 20 Hz frequency, and 5 ms pulse duration), with results given in Figure 9. When the distance between the laser head and WM is close enough (−3 mm), the highest penetration depth was achieved for both media (air and serum). Although the depth of penetration is different for various focal position values, for the −3 mm focal position the penetration depth value is almost the same, achieving the maximum value (this value was 900 μm for both media, as shown in Figure 9). As the laser radius increases, the amount of energy per unit area also increases as the distance from the focal point increases. This causes the penetration depth to increase. In addition, the largest depth of penetration was observed in serum medium for the varied focal position values compared to the air medium.

Underwater laser beam welding (ULBW) operation comprises two major steps. During the first step, two interactions occur (i.e., laser–serum and laser–metal), and the ‘beam channel’ is formed under the radiation of the incident laser beam. During the second step, the ‘beam channel’ is formed by irradiation of the laser beam on the target surface. During the welding process, the ‘beam channel’ is surrounded by a specific volume of water vapor cloud due to the evaporation of water above the molten pool [48] (therefore the highest depth of penetration was achieved in the serum medium). Moreover, the laser light is efficiently absorbed by the water, resulting in fast heating and convection movement [49]. Air has a lower absorption coefficient than water; thus, the propagation lengths in this instance are substantially longer [50].

The determination of energy, which is the most critical parameter affecting the welding depth, is one of the main steps in laser welding [50]. The relation between the penetration depth and the laser energy, by applying 5 mm/s speed, −6 mm focal position, 20 Hz frequency, and 5 ms pulse duration, are presented in Figure 10. The experimental energy values for water and air media are 19 J, 23.4 J, 25 J, and 28 J. Although, the trends in depth of penetration in accordance with energy were approximately the same, the depth of penetration in the serum medium was almost two times larger than in the air medium for all used energy inputs, as clearly shown in Figure 10. The deepest penetration was achieved using 28 J energy with penetration depths of 750 μm and 450 μm for serum and air medium, respectively. In general, larger energy input results in a higher penetration depth. With a higher energy input, the samples are heated more, and thus the welding pool becomes wider and deeper. This phenomenon causes a variation in the depth of penetration. In other words, when a higher energy is applied, a higher depth of penetration is observed. If the melting pool is too large (or too small), or if extensive vaporization occurs during welding, poor results might be achieved.

Figure 11 shows the interaction between penetration depth and SPE in both air and serum environments. For both media, the penetration force changes following the same trend. Based on the SPE, the penetration levels of the samples welded in the serum environment are higher than those in the air environment. At 44 J SPE, the maximum penetration (900 μm for both media) was recorded while the minimum penetration (225 μm for air and 380 μm for serum medium) was observed at 61.6 J SPE. It is obvious that the change in penetration depth with SPE is a fully nonlinear process. Although the SPE was at a minimum at 44 J, the highest penetration was achieved at this SPE level because focal position was −3 mm (the minimum focal position used in experiments in both media). After 129.2 J SPE, the depth of penetration increased gradually due to the increasing selected laser energy increments (from 19J to 28J).

Laser energy affects not only the depth of penetration, but also the HAZ and welding pool width [51]. Plasma absorption takes an active role at the top of the weld (on the material’s surface) where the available laser energy is the highest, resulting in a larger welding pool and HAZ width [51]. Figure 12 shows the focal position (a), energy (b), and SPE (c) versus weld width in the welding environments of air and serum. When the focal position and energy effect was investigated, other parameters were kept constant (for focal position: 5 mm/s speed, 11 J energy, 20 Hz frequency, and 5 ms pulse duration; for energy: 5 mm/s speed, −6 mm focal position, 20 Hz frequency, and 5 ms pulse duration). In the air medium, the weld width was larger than in the serum medium for different values of focal position and energy. Moreover, the same conclusion can be derived for different SPE levels. When looking at the focal position effect on penetration depth (Figure 9) and weld width (Figure 12a), the optimal focal position parameter is −3 mm, since the best penetration and weld width interaction was accomplished at this value with the formation of a keyhole. Weld width increased approximately linearly with energy for both media, as seen in Figure 12b. The correlation between SPE and weld width is also demonstrated in Figure 12c for SPE varying from 44 J to 190.4 J. For both media, the maximum (1960 μm for air and 1900 μm for serum medium) and minimum (1200 μm for air and 800 μm for serum medium) weld widths were reached at 190.4 J and 61.6 J SPE, respectively. The weld width values in both environments are quite close for the higher SPE levels (159.12 J, 170 J, and 190.4 J), since the higher energy removed water by vaporization and exhibited similar characteristics as in the air.

The hardness distributions of the welded cross-sections have been analyzed using a HARDWAY HVD-1000 AD hardness tester with a load of 300 g. The results indicated that the hardness of the WM was greater than the HAZ and the BM (380 HV (0.3)) (as reported in Ref. [7]) in both media (Figure 13). Due to the high cooling rate in LBW, a substantial increase in hardness is observed in the laser-welded Ti6Al4V alloys. In the HAZ and the welding region, rapid cooling causes martensite formation. For the titanium alloys, rapid cooling and martensitic transformation have been proven as efficient strengthening techniques [52]. Since the cooling rate in the serum environment is greater than in the air environment, the hardness of welded samples was often higher in the serum environment than in the air environment. Figure 13a represents the effect of the focal position variation on the hardness of the WM given the constant parameters of 5 mm/s speed, 11 J energy, 20 Hz frequency, and a 5 ms pulse duration. The maximum hardness (1130HV) was achieved at −4 mm focal position in the serum medium. In the air medium, the hardness values for variable focal positions were smaller than in the serum medium, except for the −5 mm focal position which was achieved the maximum hardness for the air medium, at 795 HV. To investigate the influence of the energy on hardness in air and serum media, 5 mm/s speed, −6 mm focal location, 20 Hz frequency, and 5 ms pulse length were used as the constant values with 19 J, 23.4 J, 25 J, and 28 J energy values, as shown Figure 13b. The hardness in serum media was always greater than the hardness in the air medium for investigated energy levels. At 19 J energy, the maximum value of 1410 HV was obtained in air media. Hardness values in the serum medium reduced dramatically as energy levels increased, and the lowest hardness (815 HV) was recorded in the highest energy level (28 J). In the air environment, between 19 J and 25 J laser energy, hardness linearly increased from 700 HV to 840 HV. However, it decreased to 680 HV at 28 J energy. The relation between SPE and hardness is observed in Figure 13c for the air and serum environments. In the air medium, the hardness values ranged from 600 HV to 840 HV, while in the serum media they ranged from 725 to 1410 for SPE values between 44 J and 190.4 J. For different SPE levels, the serum medium range is smaller than in the air medium; however, there is no visible correlation between hardness and SPE. To summarize, the cooling rate affects the martensite structure of the material. Since the cooling rate in the water environment is greater than the cooling rate in the air environment, the formation of martensite tends to change; therefore, the hardness will be higher in the aquatic environment.

## 4. Conclusions

In this paper, a detailed experimental investigation into laser welding of titanium alloy (Ti6Al4V) has been performed by using a Nd:YAG laser in two different welding environments: air and serum. The important laser parameters, i.e., the focal position and the laser energy, have been examined in detail. Based on the obtained results, the following conclusions can be derived:To achieve optimum penetration, the focal position should be closer to the thickness of the workpiece. (In our study, a penetration of 900 μm was obtained with the −3 mm focal position when 11 J of laser energy was applied in both investigated media—air and serum.)In the serum medium, the maximum value of hardness (1130 HV) was obtained at −4 mm focal position. In the air medium, the maximum hardness value (795 HV) was obtained for the focal position of −5 mm.In the air medium, the minimum and maximum obtained weld widths were 1200 μm and 1960 μm, respectively. In the serum medium, the minimum and maximum obtained weld widths were 800 μm and 1900 μm, respectively.When compared to the air medium, the serum medium has a higher penetration depth. The presence of salt particles in the serum may affect the process of transmitting energy from the laser to the sample.In both media, there is no visible correlation between SPE and depth of penetration and SPE and hardness. On the other hand, the SPE increases with welding width.The focal position has no significant influence on the hardness of the WM in open-air and underwater welding.While the laser energy does not exhibit a noticeable influence on hardness in air welding, increasing the energy causes a remarkable reduction in the hardness level of WM in UWW.In the weld zone and the HAZ, different forms of martensite have been observed. In UWW, the needle-shaped and the basket-shaped austenite formed in the weld zone, which caused an increase in the hardness to the point that cracks formed.

To conclude, our results show that the laser welding of Ti6Al4V titanium alloy in serum medium can be used as a promising welding technique in body fluids (e.g., in future in vivo studies), provided that it is adequately performed and the proper laser parameters are applied. To the best of our knowledge, the laser welding of Ti6Al4V titanium alloy in serum medium has not been previously reported. The experimental findings we report here can provide important insights leading to a better understanding of the mechanisms underlying the laser welding process of titanium alloys in a serum medium. In addition, our results can provide useful data for new emerging techniques of laser welding of thin biomaterials, for which the experimental variables (i.e., focal position and laser energy) must be carefully chosen in order to determine the optimum depth of penetration, weld width, and hardness.

## Figures and Tables

**Figure 1 materials-15-09088-f001:**
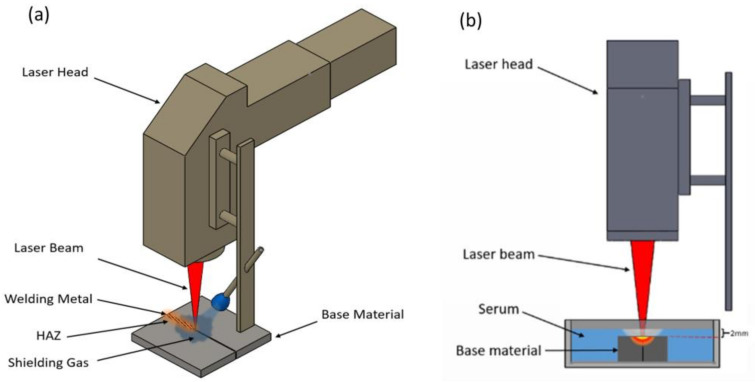
The experimental setup of the laser welding (**a**) in atmospheric air and (**b**) in a serum environment.

**Figure 2 materials-15-09088-f002:**
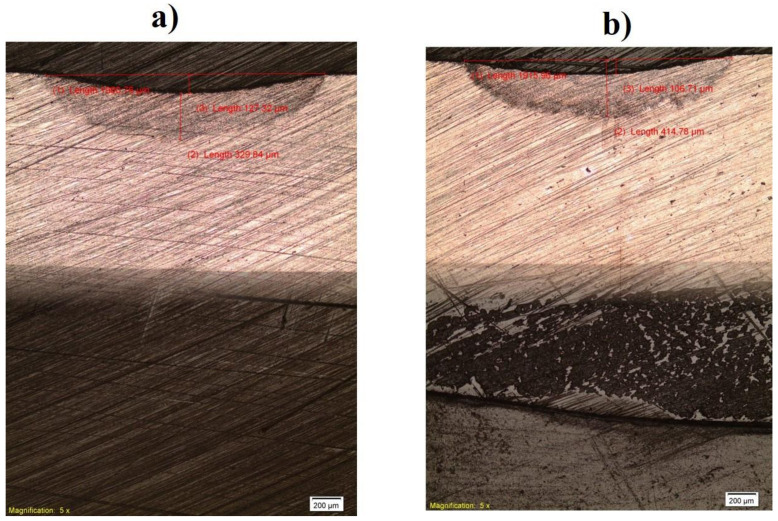
The macrographs of the experimental specimens (**a**) in air and (**b**) in serum.

**Figure 3 materials-15-09088-f003:**
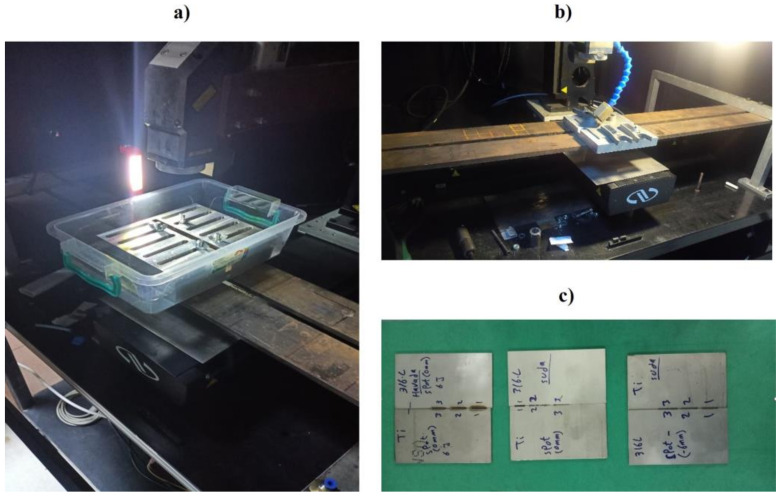
The photographs of (**a**) the experimental setup, (**b**) the welded plate, and (**c**) the samples prepared from the welded plate.

**Figure 4 materials-15-09088-f004:**
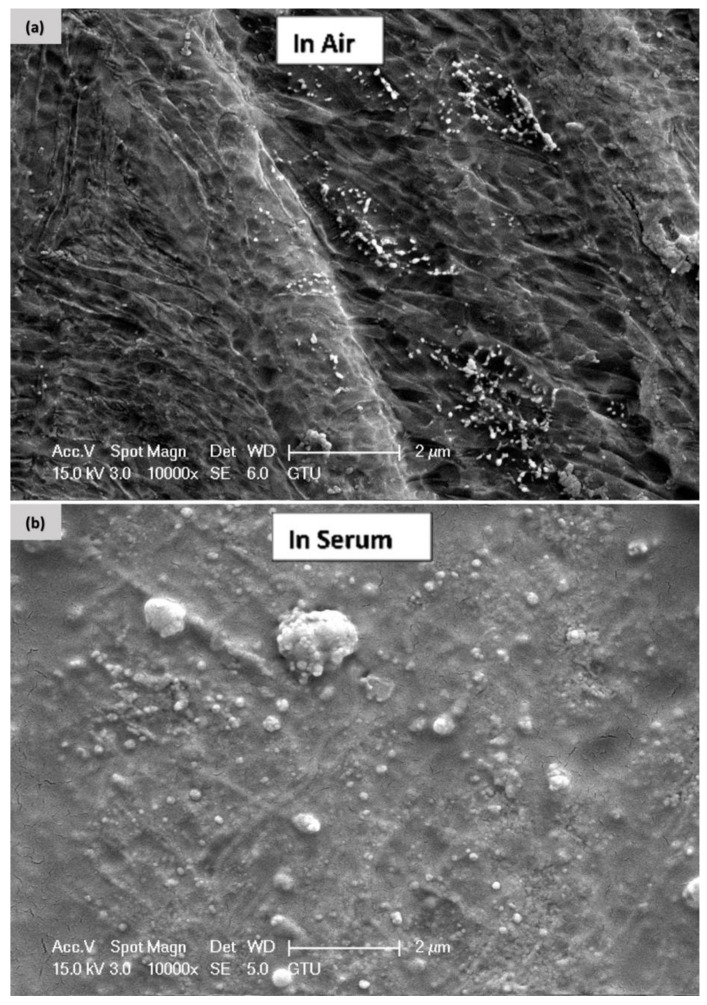
SEM micrographs of Ti6Al4V welded in air (**a**) and in serum (**b**).

**Figure 5 materials-15-09088-f005:**
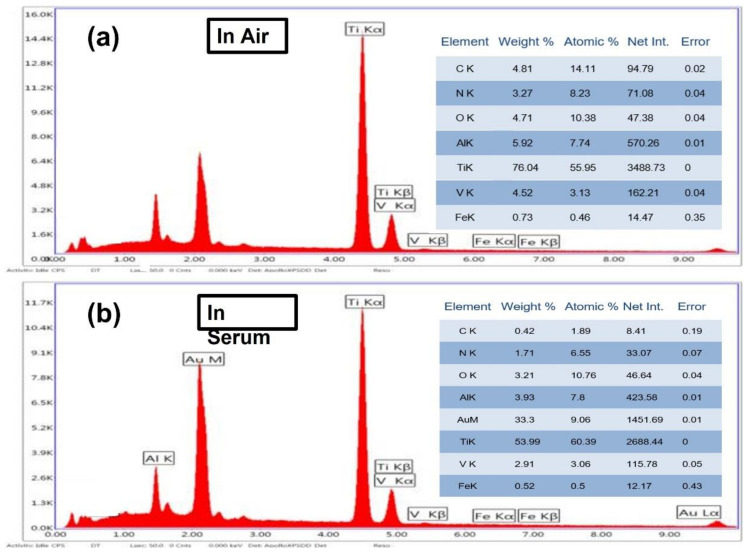
EDS spectra of Ti6Al4V welded in air (**a**) and in serum (**b**).

**Figure 6 materials-15-09088-f006:**
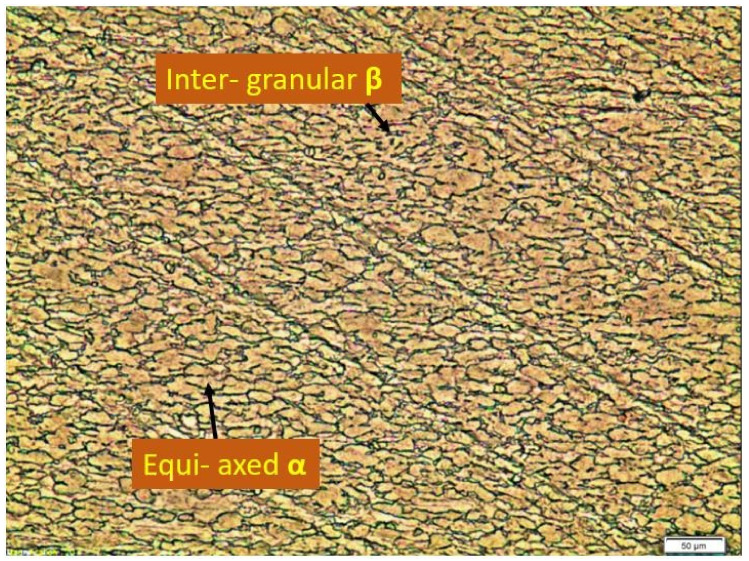
The microstructure of BM (Ti6Al4V titanium alloy).

**Figure 7 materials-15-09088-f007:**
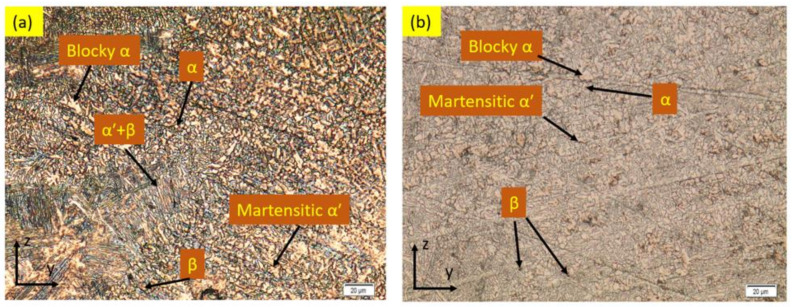
The microstructure of the fusion zones (**a**) in serum medium (**b**) in air medium.

**Figure 8 materials-15-09088-f008:**
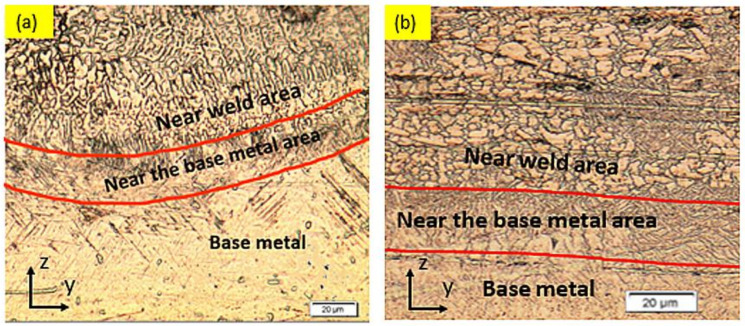
The microstructures of HAZ Ti6Al4V welded (**a**) in serum medium (**b**) in air medium.

**Figure 9 materials-15-09088-f009:**
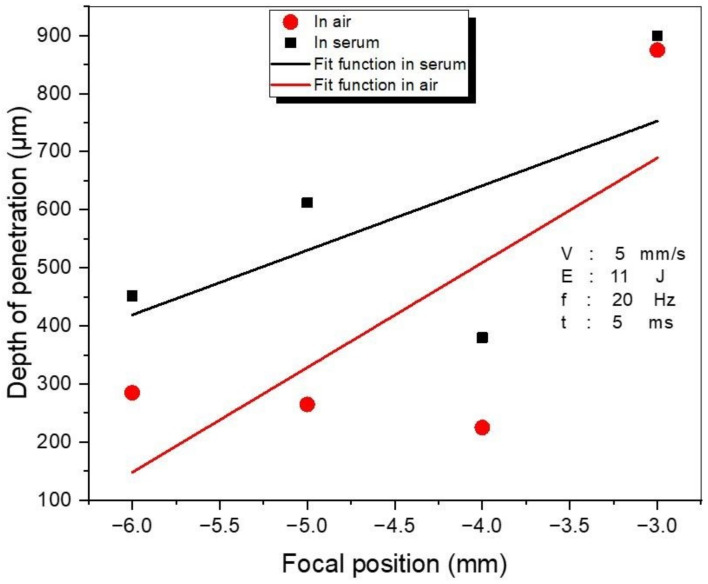
The variations in the penetration depth (μm) with focal position.

**Figure 10 materials-15-09088-f010:**
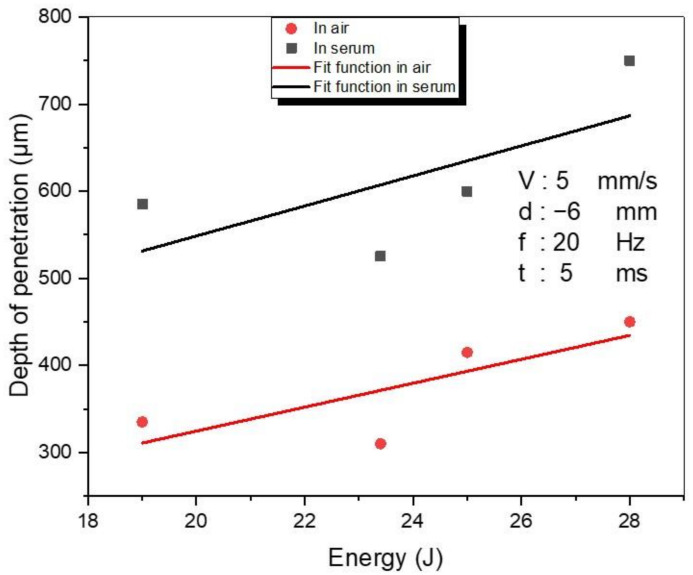
The variations in the penetration depth (μm) with energy.

**Figure 11 materials-15-09088-f011:**
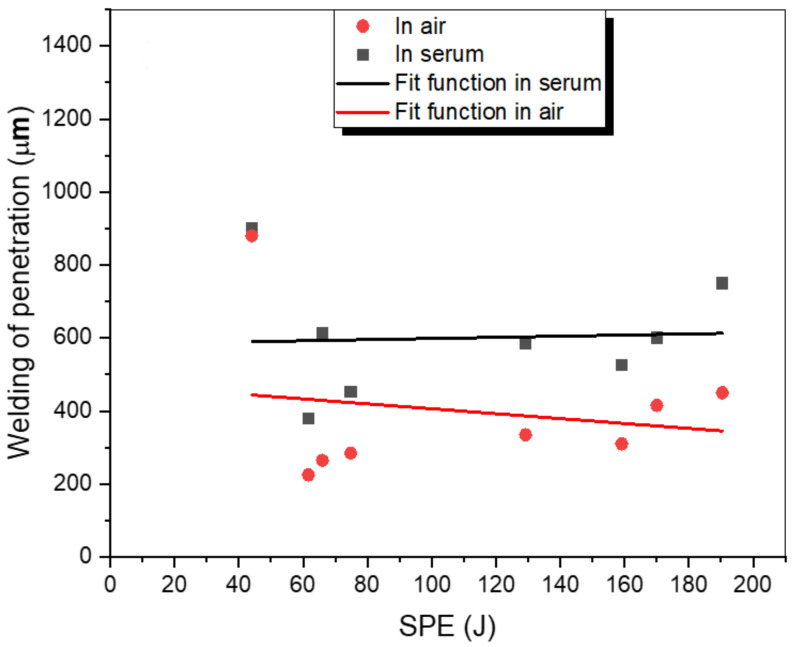
The variations in the penetration depth (μm) with SPE.

**Figure 12 materials-15-09088-f012:**
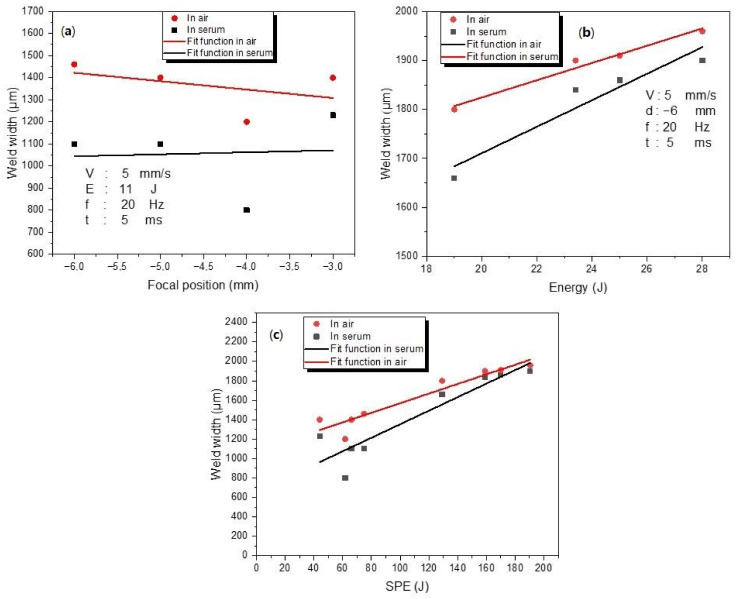
The variations in the weld width (μm) with focal position (**a**)**,** energy (**b**), and SPE (**c**).

**Figure 13 materials-15-09088-f013:**
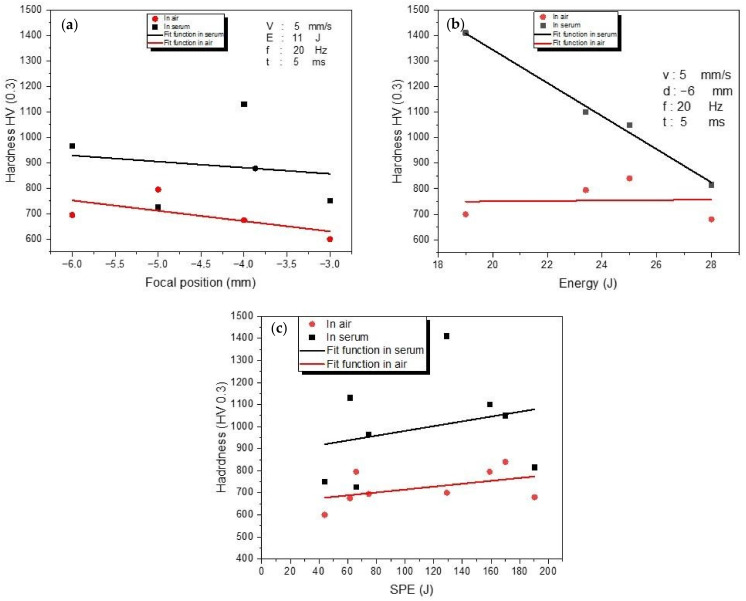
The variations of the hardness HV (0.3) with focal position (**a**), energy (**b**), and (**c**) SPE.

**Table 1 materials-15-09088-t001:** The chemical composition and mechanical and physical properties of Ti6Al4V [40].

Chemical Composition (wt.%)								
Sample	Al	V	Fe	O	C	N	H	Ti
Ti6Al4V	6	4	0.3	0.2	0.08	0.05	0.01	Bal
Basic mechanical and physical properties								
Hardness (HRC)	Density (g/cm3)		Modulus (MPa)	Elongation (%)		Tensile strength (MPa)	Thermal conductivity (W/mK)	
36	4.43		910	0.7		1000	7.3	

**Table 2 materials-15-09088-t002:** Laser parameters used in the experiments. (Argon was used for air welding as the shielding gas.)

Experiment Run	Variable			Constant			
	Energy Density (Heat Input)	Laser Power	Focal Position	Frequency	Gas Flow (Argon)	Pulse Duration	Welding Speed
	E/l	P	Z	F		t	V
	[J/mm]	[W]	[mm]	[Hz]	[l/min]	[ms]	[mm/s]
1	110	220	−3	20	18	5	5
2	110	220	−4				
3	110	220	−5				
4	110	220	−6				
5	190	380	−6				
6	230	460	−6				
7	250	500	−6				
8	280	560	−6				

## Data Availability

Data sharing is not applicable for this paper.

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
