# Peer review of "Laser Welding of Ti6Al4V Titanium Alloy in Air and a Water Medium"

_materials, 2022, doi:10.3390/ma15249088_

Round 1

Reviewer 1 Report

Dear Authors,  

I have reviewed your paper titled: "Laser Welding of Ti6Al4V Titanium Alloy in Air and a Water Mediu".

The paper fulfils the aims and scope of Materials journal, and can be considered for potential publication. However, it needs some improvements. I have some minor suggestions, which are listed below. 

General remarks:

- Please add the quantitative results into the abstract.

- You have presented 58 references. Only 10 of them have been published in last three years. I suggest to support your work with newly published references more. It will increase the visibility of your work in scintific databases. E.g., only about underwater welding, there are couple of papers this year: https://scholar.google.com/scholar?hl=pl&as_sdt=0%2C5&as_ylo=2022&q=%22wet+welding%22&btnG=

Introduction:

- "local dry cavity welding" - this is also named simply as "local cavity welding".

- molecular hydrogen - problem in UW is "diffusible hydrogen", which is responsible e.g., for cold cracking.

Materials and Methods:

- Table 1 - the source of presented values is unknown. Have you tested these composition/parameters? If yes, please mark used methods and techniques. If values were taken from standard or manufacturer datas, please add relevant description.

- Table 2 - please udnerline, why these parameters were chosen.

- What was the welding depth in serum? It has crucial influence on the properties of joints.

- You have performed some standarized tests. Please mark relevant standards.

Results:

- I cannot find any section named "Discussion". I propose to add discussion here, and change name to "Results and Discussion". Moreover, you should discuss your results with other scientific papers more. It allows to underline the biggest advantages from your work. It allows to mark the novelty of studies.

- Please add macrographs of performed specimens, where whole cross-section of specimens could be observed.

- Results are presented well.

Conclusions:

- Please support conclusions with the quantitative results.

Author Response

Please find our answers in the attachment

Reviewer 2 Report

Review report: Laser Welding of Ti6Al4V Titanium Alloy in Air and a Water Medium

1.       Shorten the length of the abstract section and add only key information in abstract section.

2.       Discuss the Novelty and clear application of the work in abstract as well as in introduction section.

3.       Shorten the length of the introduction section and add key published work and try to make a bridge between current and previous published work. Add some recently published work in area of laser welding. Also why it is so popular than arc welding:  https://doi.org/10.1016/j.ijpvp.2022.104629; https://doi.org/10.1016/j.optlastec.2021.107610.

4.       How was the composition of base plate obtained? Also mention about the mechanical properties of the base plate.

5.       Provide the image of the experimental setup, welded plate and sample prepared from the welded plate.

6.       There was discussion related to effect of focal length and energy on depth of penetration but technical discussion related to variation is missing.

7.       In place of results, use results and discussion.

8.       Metallography discussion should be the first section.

9.       Try to relate the hardness variation results with microstructure evolution along weldments.

10.    Microstructure of BM should be in the experimental section or the start of the result section.

11.    Try to add SEM image instead of poor quality optical image.

12.    It is difficult to reach any conclusion from optical image like martensite or other phases. Either add references or add good-quality SE image along with EDS spectra.

13.    In Fig. 9, one region is not etched properly. 

Author Response

Please find our answers to your comments in the attachment.

Round 2

Reviewer 1 Report

Dear Authors,
Thank you for your response. Paper has been improved a lot, and can be published in this state.

Best regards,

Author Response

Thank you very much for your approve

Reviewer 2 Report

Accepted. 

Author Response

Thank you very much for your acceptance